# Stimulating Respiratory Activity Primes Anaerobically Grown *Listeria monocytogenes* for Subsequent Intracellular Infections

**DOI:** 10.3390/pathogens7040096

**Published:** 2018-12-08

**Authors:** Nathan Wallace, Erica Rinehart, Yvonne Sun

**Affiliations:** Department of Biology, University of Dayton, Dayton, OH 45469, USA; wallacen1@udayton.edu (N.W.); erinehart1@udayton.edu (E.R.)

**Keywords:** *Listeria monocytogenes*, anaerobic respiration, fumarate, intracellular infection

## Abstract

*Listeria monocytogenes (L. monocytogenes)* is a Gram-positive, enteric pathogen and the causative agent of listeriosis. During transition through the gastrointestinal tract, *L. monocytogenes* routinely encounters suboxic conditions. However, how the exposure to the low oxygen environment affects subsequent pathogenesis is not completely understood. Our lab previously reported that anaerobically grown *L. monocytogenes* exhibited an intracellular growth defect in macrophages even though the infection took place under aerobic conditions. This phenotype suggests that prior growth conditions have a prolonged effect on the outcome of subsequent intracellular infection. In this study, to further investigate the mechanisms that contribute to the compromised intracellular growth after anaerobic exposure, we hypothesized that the lack of respiratory activity under anaerobic conditions prevented anaerobically grown *L. monocytogenes* to establish subsequent intracellular growth under aerobic conditions. To test this hypothesis, respiratory activity in anaerobically grown *L. monocytogenes* was stimulated by exogenous fumarate and subsequent intracellular pathogenesis was assessed. The results showed that fumarate supplementation significantly increased the respiratory activity of anaerobically grown *L. monocytogenes* and rescued the subsequent intracellular growth defect, likely through promoting the production of listeriolysin O, phagosomal escape, and cell-cell spread. This study highlights the importance of respiratory activity in *L. monocytogenes* in modulating the outcome of subsequent intracellular infections.

## 1. Introduction

*Listeria monocytogenes* (*L. monocytogenes*) is a Gram-positive, facultative anaerobe found in a variety of environmental habitats [1,2]. When *L. monocytogenes* enters an animal host through ingestion and establishes infections as an intracellular pathogen, it can cause diseases with a high mortality rate in immunocompromised individuals. For example, within the 1998–2008 decade, confirmed *L. monocytogenes* outbreaks led to a total of 38 deaths out of 359 illnesses [3]. Moreover, it was estimated that *L. monocytogenes* infections were responsible for an annual cost burden of $2.8 billion in the United States [4]. The high mortality rate and the cost burden associated with *L. monocytogenes* infections together establish *L. monocytogenes* a critical foodborne pathogen that needs to be under routine surveillance in the food industry. *L. monocytogenes* exhibits a unique ability to survive and grow in food preservation processes that are generally considered antimicrobial, such as high osmotic stress, low pH, and refrigeration temperatures [5,6,7]. Therefore, to effectively prevent *L. monocytogenes* contamination in food products and protect high-risk individuals from *L. monocytogenes* exposure, it is necessary to consider and understand how *L. monocytogenes* responds and adapts to different environmental conditions. 

Being a facultative anaerobe, *L. monocytogenes* is able to survive and proliferate in conditions of low to no oxygen. As *L. monocytogenes* traverses the human gastrointestinal tract during infection, it encounters varying concentrations of oxygen [8,9]. How this exposure and potential adaptation to anaerobic conditions impact subsequent infections is not completely understood. *L. monocytogenes* cultured under strict anaerobic conditions has an initial invasion advantage compared to those cultured under aerobic condition, a phenotype mediated through the upregulating the *Listeria* adhesion protein (LAP) under anaerobic conditions [10,11]. We have also observed this initial invasion advantage in anaerobically grown *L. monocytogenes* during infections of different cell lines. However, over a period of 8 hours, prior anaerobic exposure actually led to an intracellular growth defect [12]. Because the infections were performed in the presence of oxygen, this observation suggests that anaerobic exposure exerts a sustained effect on subsequent infections beyond initial invasion.

*L. monocytogenes* adaptation to anaerobic conditions includes shifts in carbon metabolism. In defined medium using glucose as the sole carbon source, aerobic growth results in the production lactate, acetate, and acetoin, whereas anaerobic growth results in the production of lactate with minor accumulation of acetate, formate, and ethanol [13]. In rich BHI medium, acetoin production was observed under aerobic but not anaerobic conditions [12]. Although *L. monocytogenes* is primarily a lactic acid fermenter, it contains genes coding for a fumarate reductase. The gene *lmo0355*, which encodes a subunit in fumarate reductase in strain EGDe, is upregulated under anaerobic conditions and allows *L. monocytogenes* an enhanced anaerobic growth in the presence of fumarate [14]. Therefore, it is likely that *L. monocytogenes* is capable of carrying out anaerobic respiration using fumarate as the terminal electron acceptor in the absence of oxygen.

To further understand the impact of anaerobic adaptation on *L. monocytogenes* pathogenesis, we tested whether the defective intracellular growth after anaerobic exposure was caused by anaerobically grown *L. monocytogenes* having reduced respiratory activity and, thus, a delay in aerobic growth. If the lack of respiratory activity contributes to the intracellular growth defect in anaerobically grown *L. monocytogenes*, stimulating anaerobic respiratory activity using an alternative electron acceptor such as fumarate should rescue the intracellular growth defect.

## 2. Results

### 2.1. Fumarate Supplementation Results in Enhanced Anaerobic Respiratory Activity

To confirm that supplementation of fumarate can indeed stimulate respiratory activity in *L. monocytogenes* under anaerobic conditions, multiple in vitro assays investigating the physiological responses to fumarate were performed. During anaerobic growth in BHI at 37 °C, supplementation of 50 mM fumarate resulted in a significant increase in overnight culture optical density (OD), compared to no fumarate controls (Figure 1a). This anaerobic, fumarate-induced increase in optical density is accompanied by a significant increase in culture pH (Figure 1b) and acetoin level (Figure 1c), a significantly higher tetrazolium reduction activity (Figure 1d), and a significantly higher intracellular ATP levels (Figure 1e). In contrast, supplementation of fumarate under aerobic conditions did not significantly alter culture OD, pH, acetoin level, or tetrazolium reduction activity (Figure 1a–d) but resulted in a significantly decreased intracellular ATP levels (Figure 1e) compared to aerobic, no fumarate controls. Together, these data indicate that fumarate supplementation results in enhanced anaerobic respiratory activity.

### 2.2. Fumarate Supplementation Enhanced Transition from Anaerobic to Aerobic Growth

Next, to determine whether the lack of respiratory activity in anaerobically grown *L. monocytogenes* was the cause of subsequent defect in aerobic growth, the effects of fumarate supplementation on transition from anaerobic to aerobic growth in vitro as well as inside macrophages were tested. Aerobically or anaerobically grown *L. monocytogenes* was used to inoculate fresh BHI and the resulting cultures were incubated at 37 °C under aerobic conditions. Compared to aerobically-grown *L. monocytogenes*, anaerobically grown *L. monocytogenes* exhibited a notable decrease in aerobic growth (Figure 2a), a phenotype demonstrating in vitro growth defects during anaerobic to aerobic transition. In comparison, anaerobically grown, fumarate-stimulated *L. monocytogenes* exhibited a faster adaptation to aerobic growth (Figure 2a). Similar phenotypes were also observed during transition into intracellular growth. Overnight *L. monocytogenes* cultures were used to infect RAW264.7 macrophages for 30 min at a multiplicity of infection (MOI) of 10 and assayed for intracellular growth under aerobic incubation. The extent of intracellular growth was similar in macrophages infected with aerobically grown *L. monocytogenes* with or without prior fumarate stimulation (Figure 2b). However, in macrophages infected with anaerobically grown *L. monocytogenes*, the intracellular growth was greatly enhanced by exposing bacteria to fumarate prior to infection (Figure 2b). These in vitro and intracellular growth characteristics demonstrate that fumarate-stimulated *L. monocytogenes* exhibits a faster anaerobic to aerobic growth transition in vitro and inside host cells.

### 2.3. Sublethal Levels of CCCP Compromise Respiratory Activity In Aerobically Grown L. monocytogenes

If respiratory activity prior to infection is key in subsequent intracellular success, compromising respiratory activity in aerobically grown *L. monocytogenes* should, in theory, result in reduced pathogenesis. To test this hypothesis, *L. monocytogenes* was grown in BHI at 37 °C in the presence of the proton ionophore, CCCP, to reduce its respiratory activity and its subsequent infections were assessed. A working concentration of 1 µM CCCP was established to reduce aerobic growth to a level similar to that of anaerobic growth without complete growth inhibition. Compared to no supplement, both aerobic and anaerobic growth with 1 µM CCCP was significantly reduced (Figure 3a), however there was a greater growth inhibition in the aerobically grown *L. monocytogenes* compared to that in anaerobically grown *L. monocytogenes*. Compared to no supplementation, CCCP treatment resulted in significantly reduced culture pH and tetrazolium reduction activity under aerobic, but not anaerobic, conditions (Figure 3b,c). *L. monocytogenes* grown with or without CCCP treatment was then used to infect RAW264.7 macrophages to assess the impact of prior CCCP treatment on subsequent intracellular growth. At one hour post infection (hpi), while prior CCCP treatment did not affect subsequent infections by anaerobically grown *L. monocytogenes*, there was a significantly higher intracellular CFU in macrophages infected with the aerobically grown, CCCP-treated *L. monocytogenes* than those infected with aerobically grown, untreated controls. However, there was no difference in intracellular growth over an eight-hour period for bacteria grown with or without CCCP (data not shown).

### 2.4. Modulations of Respiratory Activity Alter Production of LLO and Actin Co-Localization

To establish a successful intracellular life cycle, *L. monocytogenes* needs to escape the phagosome and enter the host cytoplasm to grow. This phagosomal escape is mediated by the secreted virulence factor, listeriolysin O (LLO). Therefore, to determine whether the effects of prior exposure to fumarate and CCCP on subsequent infections were attributed to different levels of LLO production, the effects of fumarate and CCCP on LLO production in vitro were investigated. The activity of the secreted LLO, when normalized by culture optical density, was significantly higher in aerobic culture supernatant than in anaerobic culture supernatant (Figure 4a). Supplementation of fumarate resulted in significantly increased supernatant activity in aerobically or anaerobically grown cultures (Figure 4a). Supplementation of CCCP, in contrast, resulted in significantly decreased LLO activity in aerobic but not anaerobic cultures, compared to no CCCP controls. (Figure 4b). To confirm supernatant LLO activity is representative of protein abundance, the secreted LLO protein abundance was quantified using immunoblotting (Figure 4c, left) followed by pixel density analysis (Figure 4c, right). Compared to no fumarate controls, the abundance of LLO protein was significantly increased, under both aerobic and anaerobic conditions, in cultures stimulated with fumarate. However, CCCP treatment led to no significant difference in LLO abundance compared to no CCCP controls.

To further investigate whether supplementations of fumarate and CCCP in the growth of the inoculum would impact subsequent phagosomal escape, bacterial actin co-localization assay was performed. No actin co-localization was observed at 2 hpi in macrophages infected by anaerobically grown *L. monocytogenes*. However, in macrophages infected by anaerobically grown, fumarate-stimulated *L. monocytogenes*, an elevated 26.7 ± 6.8% actin co-localization was observed (Figure 4d), a phenotype consistent with the positive effects of fumarate on LLO production (Figure 4c). However, fumarate supplementation in aerobically grown *L. monocytogenes* prior to infection resulted in a significantly reduced level of actin co-localization compared to untreated samples (Figure 4d). Additionally, while no actin co-localization was observed in macrophages infected with anaerobically grown *L. monocytogenes* regardless of the presence or absence of prior CCCP treatment, CCCP treatment led to a significant decrease in the actin co-localization in macrophages infected with aerobically grown *L. monocytogenes* (Figure 4e). These results show that manipulations of respiratory activity with fumarate or CCCP can lead to subsequent changes in virulence factor production and infection.

### 2.5. Prior Fumarate Treatments Enhance Cell-Cell Spread 

To further understand the sustained effects of *L. monocytogenes* respiratory activity prior to host cell entry on long-term infections, plaque assays were performed with bacteria grown aerobically or anaerobically with or without fumarate or CCCP treatments. Plaque diameters were measured at three days post infection as a proxy for cell-to-cell spread. In cells infected with fumarate-stimulated *L. monocytogenes* grown under either aerobic or anaerobic conditions, plaque sizes were significantly larger than those from cells infected with *L. monocytogenes* without fumarate supplementation (Figure 5a). In contrast, CCCP treatment in aerobically, but not anaerobically grown, *L. monocytogenes* resulted in a significant decrease in plaque size (Figure 5a). It is important to note that during the three days of infection, neither fumarate nor CCCP was added to the cell culture media. Therefore, the differences in plaque sizes reflect a long-lasting impact of prior fumarate or CCCP exposure on subsequent infections.

To investigate whether fumarate exposure by host cells prior to infection also affected subsequent infections, plaque sizes were compared between infections where either the fibroblasts or both *L. monocytogenes* and fibroblasts were pre-treated with fumarate prior to infections. No fumarate was added during infection to better examine the long-term effects of prior fumarate exposure. In fumarate-treated fibroblasts, plaque sizes were significantly larger than those in non-treated fibroblasts (black bars, no fumarate controls; white bars, fumarate-treated fibroblasts; Figure 5b). This increase was further enhanced if *L. monocytogenes* was pre-treated with fumarate (gray bars, Figure 5b). For anaerobically grown *L. monocytogenes*, in particular, fumarate pre-treatments in both *L. monocytogenes* and fibroblasts resulted in complete clearing with undeterminable plaque size (CL, Figure 5b). These results confirm that manipulations of respiratory activities with fumarate or CCCP in *L. monocytogenes* prior to infections can dramatically impact subsequent infection outcomes with observable effects for as long as three days post infections. Moreover, fumarate treatment in fibroblasts also resulted in greatly enhanced infections, particularly for anaerobically grown *L. monocytogenes*.

## 3. Discussion

This study investigated the role of respiratory activity in *L. monocytogenes* in modulating the outcome of subsequent infections. Exogenous fumarate was added as a terminal electron acceptor to stimulate anaerobic respiration while the proton ionophore, CCCP, was added to reduce aerobic respiratory activity. Based on in vitro and intracellular growth characterizations, anaerobic, fumarate-stimulated *L. monocytogenes* exhibits increased respiratory activity, allowing for faster anaerobic to aerobic growth transition and a subsequent enhancement in pathogenesis. Conversely, aerobic, CCCP-treated *L. monocytogenes* exhibits reduced respiratory activity, phagosomal escape, and cell-cell spread. Because fumarate or CCCP treatments took place during the growth of *L. monocytogenes* inoculum or in fibroblasts, and were never added during infections, these results highlight the respiratory activity in *L. monocytogenes* prior to infection as a key determinant in infection outcome.

*L. monocytogenes* is a facultative anaerobe and is exposed to suboxic to anoxic environments during its transmission and host colonization. How this exposure to fluctuating levels of oxygen affects subsequent infections is not completely understood. It has been shown that anaerobic growth prior to infection resulted in a significant increase in invasion [10,11,12]. This initial success in entry into host cells could be recapitulated in aerobically grown bacteria by reducing the respiratory activity with CCCP (Figure 3d). However, the invasion advantage of anaerobically grown *L. monocytogenes* did not extend into long-term intracellular growth. In fact, *L. monocytogenes* that was grown anaerobically exhibited a significantly compromised intracellular growth compared to those grown aerobically prior to infection [12]. It is important to note that these cell culture-based infections lasted 8 h under aerobic incubation. Therefore, the defect observed at 8 (Figure 2b) or 72 h post infection (Figure 5a) suggests that prior adaptation to anaerobic conditions severely compromises the ability of *L. monocytogenes* to establish a productive, intracellular infection under aerobic conditions. However, stimulation of respiratory activity under anaerobic conditions, as demonstrated by the use of fumarate, can successfully rescue the subsequent intracellular growth defect.

The choice for fumarate in our study was based mainly on the predicted ability of *L. monocytogenes* to utilize fumarate as an alternative electron acceptor in the absence of oxygen [15,16]. However, the extent by which fumarate is involved in *L. monocytogenes* growth and pathogenesis goes beyond anaerobic respiration. Supplementation of fumarate as an intermediate in the tricarboxylic acid (TCA) cycle, as reported here (Figure 4a–c) and elsewhere [12], led to a significant increase in *L. monocytogenes* LLO production under both aerobic and anaerobic conditions. Therefore, it is possible that the increase in LLO production in anaerobic, fumarate-treated *L. monocytogenes* was a result of both stimulating TCA cycle and anaerobic respiratory activities. Interestingly, while LLO production was increased in both aerobically and anaerobically grown *L. monocytogenes* by fumarate treatment, actin colocalization was only increased in macrophages infected by fumarate-treated, anaerobically grown *L. monocytogenes*. In macrophages infected by aerobically grown *L. monocytogenes*, prior fumarate treatment resulted in a decreased level of actin colocalization. First, in vitro LLO production is not the absolute predictor for phagosomal escape, especially when LLO production was measured using LLO accumulated in vitro over 16–20 h of overnight culturing. Second, it is possible that while fumarate supplementation enhances LLO production in both aerobically and anaerobically grown *L. monocytogenes*, its effect on subsequent intracellular ActA production is not equivalent between aerobic or anaerobically grown *L. monocytogenes*. Given that aerobically grown, fumarate-treated bacteria exhibited no defects in intracellular growth compared to aerobically grown, no fumarate controls (Figure 2b), it is likely that the decrease in actin colocalization is not a result of defects in phagosomal escape but in ActA production in the cytosol. How respiratory activity prior to infections affects subsequent intracellular ActA production is not known and is under current investigation.

The enhancing effect of exogenous fumarate on *L. monocytogenes* pathogenesis reported here raises concerns over the use of fumarate as a food preservative. The United States Food and Drug Administration recommends fumaric acid and salts of fumaric acid as safe to be used in food for direct human consumption “at a level not in excess of the amount reasonably required to accomplish the intended effect” [17]. As a result, fumarate and its acid form, fumaric acid, have been tested as an antimicrobial additive in a variety of food products. For example, addition of fumaric acid (1%) to the surface of lean beef inoculated with *L. monocytogenes*, followed by 5 s of incubation at 55 °C, resulted in a 1-log reduction in CFU [18]. Fumaric acid (0.25%) also caused a complete inhibition of planktonic *L. monocytogenes* [19]. Therefore, co-occurrence of fumarate and suboxic packaging might create an environment where *L. monocytogenes* can persist and develop an increased capability for subsequent intracellular growth. More research is needed to better understand the long-term impact of fumarate in food storage on the behavior of *L. monocytogenes* cells that have survived the antimicrobial effects of fumarate.

Based on the enhanced cell-cell spread in fumarate-treated fibroblasts (Figure 5b), the potential exposure to elevated fumarate by host cells might also increase the severity of *L. monocytogenes* infections. Curiously, when *L. monocytogenes* and fibroblasts were both treated with fumarate prior to infection, the resulted enhancement in plaque size was much more pronounced in cells infected with anaerobically grown *L. monocytogenes* compared to those infected with aerobically grown *L. monocytogenes* (Figure 5b). This observation suggests that exposure to fumarate prior to infections might contribute to a different host response to aerobically versus anaerobically grown *L. monocytogenes*. Therefore, we consider here possible scenarios for elevated fumarate exposure to host cells. Although fumarate is typically generated and consumed as part of the TCA cycle pathway located inside mitochondria, a cytosolic isoform of fumarate hydratase (FH), which catalyzes the hydration of fumarate to malate, has been described [20]. In fact, individuals with FH deficiency exhibit severe neurological and developmental diseases [21]. Moreover, the oncogenic effects of fumarate accumulated in FH-deficient cells have helped introduce fumarate as a potential oncometabolite [22]. Elevated levels of fumarate in FH-deficient cells also lead to succination of glutathione, resulting in increased production of reactive oxygen species [23] and defects in respiratory chain activities [24]—both processes may implicate the outcome of *L. monocytogenes* infections [25,26]. Whether exogenous fumarate supplementation to fibroblasts in our experimental design leads to an increase in cytosolic fumarate or similar responses in ROS production or mitochondrial functions is not clear. However, conditions where there are elevated levels of host-derived fumarate might greatly alter *L. monocytogenes*-host interactions.

The role of anaerobic respiration in modulating bacterial pathogenesis has been demonstrated in other enteric pathogens. For example, tetrathionate respiration provides *Salmonella enterica* serovar Typhimurium with advantage to colonize the inflamed intestines [27,28]. Cholera toxin production by *Vibrio cholerae* is also enhanced by anaerobic respiration [29,30]. Recently, it was demonstrated that the ability of *L. monocytogenes* to perform extracellular electron transfer conferred a clear fitness advantage in vivo [31]. When enteric bacterial pathogens transit through the suboxic host intestinal lumen, adaptive strategies to maintain redox homeostasis can modulate virulence regulation and offer a competitive advantage amidst the endogenous fermentative organisms. Moreover, based on findings from this study, these anaerobic adaptations can sustain long into aerobic environment and alter infection outcomes. Understanding the extent of the anaerobic adaptations in the pathogens will greatly enrich our ability to provide protection from infections. 

## 4. Conclusions

This study shows that using fumarate as an alternative electron acceptor to stimulate anaerobic respiratory activity in *L. monocytogenes* prior to infections leads to significantly enhanced pathogenesis, including increased LLO production, phagosomal escape, and cell-cell spread. Moreover, the enhancement from the increased respiratory activity in anaerobically adapted *L. monocytogenes* is sustainable for more than three days under aerobic conditions, suggesting that prior growth conditions play a significant long-term role for subsequent intracellular infection outcome. Therefore, the use of fumarate as a food additive or preservative may need to be reevaluated in food products with high potential for *L. monocytogenes* contamination. Furthermore, prior growth conditions of *L. monocytogenes*, either during food storage or intestinal transit, may be critical in determining subsequent infection success.

## 5. Materials and Methods

### 5.1. Bacterial Strains and Culture Conditions

*L. monocytogenes* strain 10403s was grown from isolated colonies on freshly streaked brain–heart infusion (BHI) plates (<1 week) at 37 °C. All cultures were grown in filter-sterilized BHI media to ensure consistency. Aerobic cultures were grown with agitation at 250 RPM to ensure adequate oxygen diffusion. Anaerobic cultures were grown in a temperature-controlled incubator inside an anaerobic chamber (Coy Laboratory, Type A) with a nitrogenous atmosphere containing 2.5% hydrogen. Optical density (OD) was measured in an optically clear 96-well plate at 600 nm with a volume of 200 µL per well using a 96-well plate reader (Biotek Synergy4). Sodium fumarate (Acros Organics) was prepared as 1 M stock solution in deionized water, filter-sterilized, and added directly to the media at the desired concentration of 50 mM. Carbonyl cyanide m-chlorophenyl hydrazone (CCCP) (Alfa Aesar) was prepared as a 10 mM stock in dimethyl sulfoxide (DMSO) (Amresco, Solon, OH, USA) and added directly to the media at the desired concentration of 1 µM.

### 5.2. Intracellular Growth Curves

Murine peritoneal macrophages RAW264.7 (ATCC TIB-71) were grown in Dulbecco’s Modified Eagle Media (DMEM) (Thermo Scientific, Waltham, MA, USA) supplemented with 10% (*v*/*v*) heat inactivated fetal bovine serum (JRScientific, Woodland, CA, USA), HEPES (10 mM), and glutamine (2 mM) in a 37 °C incubator with a 5% CO_2_ atmosphere. Prior to infections, cells were seeded in a 24-well tissue culture plate and grown for 14–18 h. Overnight cultures of *L. monocytogenes* were used for infections at a MOI of 10. Bacteria diluted in cell culture medium were added to each well (500 µL) and incubated for 30 min. Following incubation, media were aspirated, and cells were washed twice with sterile Dulbecco’s phosphate-buffered saline (DPBS). Fresh media (1 mL per well) containing 10 µg/mL gentamicin stock was added to each well. To enumerate intracellular bacteria, cell culture media were aspirated off and sterile 0.1% (*v*/*v*) Triton X-100 (Fisher BP151-100) was added to each well (200 µL per well) to lyse host cells. Host cells were lysed at 1, 2, 4, and 8 h post infection (hpi). Lysates were diluted and spread on LB plates. Colonies on plates were counted using an automatic colony counter (Synbiosis aCOLyte 3) after 24–48 h of incubation in a 37 °C incubator.

### 5.3. Tetrazolium Reduction Assay

Overnight cultures of *L. monocytogenes* were washed with Dulbecco’s phosphate-buffered saline (DPBS) and normalized by OD600 nm to a final volume of 1 mL. (3-(4,5-dimethylthiazol-2-yl)-2,5-diphenyltetrazolium bromide (MTT) (Invitrogen, Carlsbad, CA, USA) working solution (50 µL, 0.5 mg/mL) was added to 50 µL of bacterial sample or blank control and incubated at 37 °C for 1 h in a 96-well flat-bottomed plate. Following incubation, 100 µL of DMSO was added to each sample and the plate was placed in a 37 °C incubator shaking for 15 min. Following incubation, samples were read at 540 nm to quantify the level of MTT reduction.

### 5.4. Measurement of Supernatant Acetoin

The Voges–Proskauer test was adapted to quantify acetoin production in the supernatant of overnight *L. monocytogenes* cultures. Supernatant or acetoin standards (100 µL) were placed into a sterile micro-centrifuge tube followed by additions of 70 µL of 0.5% creatine monohydrate (Sigma, Kawasaki, Japan) in water, 100 µL of 5% 1-Napthol (Sigma) in water, and 100 µL of 40% KOH (Chempure, Plymouth, MI, USA) in 95% EtOH. Samples were incubated at room temperature for 15 min and the absorbance was read at 560 nm. A standard curve was constructed to calculate the concentration of acetoin in culture supernatant samples.

### 5.5. LLO Hemolytic Assay

Hemolytic assays were performed using supernatant from overnight cultures to measure the activity of secreted listeriolysin O (LLO). Each sample (100 µL) was incubated at room temperature with dithiothreitol (DTT) (Amresco) (5 µL, 0.1 M) for 15 min. A positive control (0.1% Triton X-100) and a negative control (blank BHI media) were included for each experiment. After incubation, samples were serially diluted using hemolysis buffer containing: dibasic sodium phosphate (35 mM) and sodium chloride (125 mM) brought to pH 5.5 with acetic acid. Defibrinated sheep’s blood (Hemostat Laboratories) was diluted to a hematocrit of 2% and then added to each sample for a final hematocrit of 1%. Samples were incubated at 37 °C for 30 min. After incubation, all samples were spun down at 2000 rpm for 5 min to pellet intact blood cells. Supernatant lysate (120 µL) was transferred to a flat bottom 96-well plate for OD measurement at 541 nm as an indicator of LLO activity. Hemolytic unit was calculated as the inverse of the dilution factor at which half complete lysis using 0.1% Triton X-100 occurred and subsequently normalized with original culture OD measured at absorbance at 600 nm. Samples that did not produce lysis at a level more than half of complete lysis were designated as “below detection” for their hemolytic units. Supernatant samples from anaerobic cultures typically generate activities at or slightly above “below detection” levels.

### 5.6. SDS-PAGE and LLO Immunoblotting

Overnight cultures of *L. monocytogenes* were used to perform SDS-PAGE and immunoblotting analysis. Cultures were first normalized to OD measured at 600 nm using BHI media and centrifuged at 10,000 rpm for 3 min to separate supernatant from bacterial pellets. Supernatant samples were precipitated with 10% trichloroacetic acid at 4 °C for 1 h. Following precipitation, samples were centrifuged at 15,000 rpm for 15 min at 4 °C. Cold (500 µL, ~4 °C) acetone was added to each sample pellet and centrifuged for 15 min at 4 °C. Supernatant was discarded and pellets were allowed to air dry for at least 5 min. The resulting pellets from TCA precipitation were resuspended in 12 µL of 2x sample buffer and heated at 95 °C for 5 min prior to separation by SDS-PAGE. Following separation, proteins were transferred to methanol-treated polyvinylidene difluoride (PVDF) membrane using a semi-dry transfer apparatus for 30 min at a constant 20v. Membranes were blocked in a 10 mL solution of 1X Tris buffered saline (TBS), 3% instant non-fat dry milk, and 0.05% TWEEN20 (Fisher Scientific, Hampton, NH, USA) for 20 min at room temperature on a rotating platform. Following blocking, membranes were placed into a 10 mL solution of 1X TBS, 3% milk, 0.05% TWEEN20, and anti-LLO rabbit primary antibody (1:10,000, abcam^®^ ab43018). Membranes were incubated overnight at 4 °C on a rotating platform. Following overnight incubation in primary antibody, the membranes were placed in a 10 mL solution of 1X TBS, 3% milk, 0.05% TWEEN20, and secondary goat-anti-rabbit HRP antibody (1:10,000, abcam^®^ ab6721). Membranes were incubated in secondary antibody for at least 90 min at room temperature on a rotating platform. Following incubation in secondary antibody, the membranes were washed with 1X TBS with 0.05% TWEEN20 three times with quick rinses in double deionized water. Excess water was dripped off of the membranes and 1 mL of HRP chemiluminescent substrate was added to each membrane. Membranes were placed into a film development cassette with classic autoratiography film (MidSci, Valley Park, MO, USA). Developed films were scanned and pixel density analysis was performed using ImageJ.

### 5.7. Actin Co-Localization by Immunofluorescence Microscopy

RAW264.7 macrophages were plated onto sterile coverslips (18 by 18 mm) inside six-well plates containing DMEM with 10% FBS at 1 million cells per well one day prior to infections. *L. monocytogenes* overnight cultures were washed twice in DPBS and diluted in cell culture media for infection at a MOI of 10. At 2 h post infection (hpi), media was aspirated and coverslips were fixed in paraformaldehyde (3.7% in PBS) overnight at 4 °C. For immunofluorescence microscopy, each coverslip was washed with TBS-T (25 mM Tris-HCl, 150 mM NaCl, 0.1% Triton X-100) and blocked with TBS-T with 1% bovine serum albumin (BSA). Anti-*L. monocytogenes* serum (1:500 in TBS-T with 1% BSA; Thermo Scientific PA1-30487) was added onto each coverslip and incubated at room temperature overnight. Each coverslip was washed in 5 mL of TBS-T prior to incubation with secondary antibodies, phalloidin-iFluor 594 (1:400, abcam ab176757) and AlexaFluor 488-goat anti-rabbit antibody (1:400, abcam ab150077), in TBS-T with 1% BSA. One hundred intracellular bacteria per experimental replicate were scored visually in randomly chosen fields using a Nikon fluorescence microscope for the presence or absence of actin clouds.

### 5.8. Quantification of Intracellular ATP

Intracellular ATP levels were quantified using a Molecular Probes™ ATP determination kit. Samples were prepared according to manufacturer’s suggested protocol. Overnight cultures of *L. monocytogenes* were grow aerobically, anaerobically, with or without fumarate were washed and resuspended with PBS to a final concentration of 10^8^ CFU/mL based on culture optical density measured at 600 nm. Washed bacterial cells were incubated in a dry heating block at 100 °C for 5 min to release intracellular ATP. Samples were then centrifuged briefly at 10,000 RPM to pellet any intact cells. Supernatant from each sample (10 µL) was placed in a black 96-well plate with 90 µL of reaction master mix (double deionized H_2_O reaction buffer, DTT, D-Luciferin, and firefly luciferase as described in manufacturer’s protocol). Bioluminescence was measured by a BioTek Synergy 4 plate reader.

### 5.9. Fibroblast Plaque Assay

Murine fibroblast cells (ATCC CRL-2648) were seeded in six-well plates and allowed to form a monolayer for 72 h. Where necessary, fibroblast cells were seeded with the addition of 50 mM fumarate during the 72 h incubation. On day of infection, 1 mL of overnight *L. monocytogenes* cultures were washed and resuspended in 100 µL of DPBS. Monolayers were washed three times with DPBS and were infected with 1 mL per well of fresh DMEM containing 6 µL of *L. monocytogenes* suspension. At 1 hpi, the cells were covered with a 3 mL mixture of DMEM containing 10 µg/mL gentamicin and 0.7% agarose. Plates were incubated at 37 °C in a 5% CO_2_ incubator for 72 h. Following incubation, wells were stained with 1 mL of a neutral red solution (0.33% (*wt*/*vol*) in DMEM) for one hour followed by PBS washes. Plaque diameters were measured using the ruler function in GIMP software.

### 5.10. Statistical Analysis

All results were analyzed and graphed using Microsoft Excel (Microsoft Corporation, Redmond, WA, USA). Statistical significance was determined using a Student’s two-tailed T-test for triplicate experiments representative of three separate experiments and error bars on these experiments represent standard deviation. For experiments with n ≥ 9 a single factor ANOVA analysis was performed with error bars representing standard error of the mean (SEM).

## Figures and Tables

**Figure 1 pathogens-07-00096-f001:**
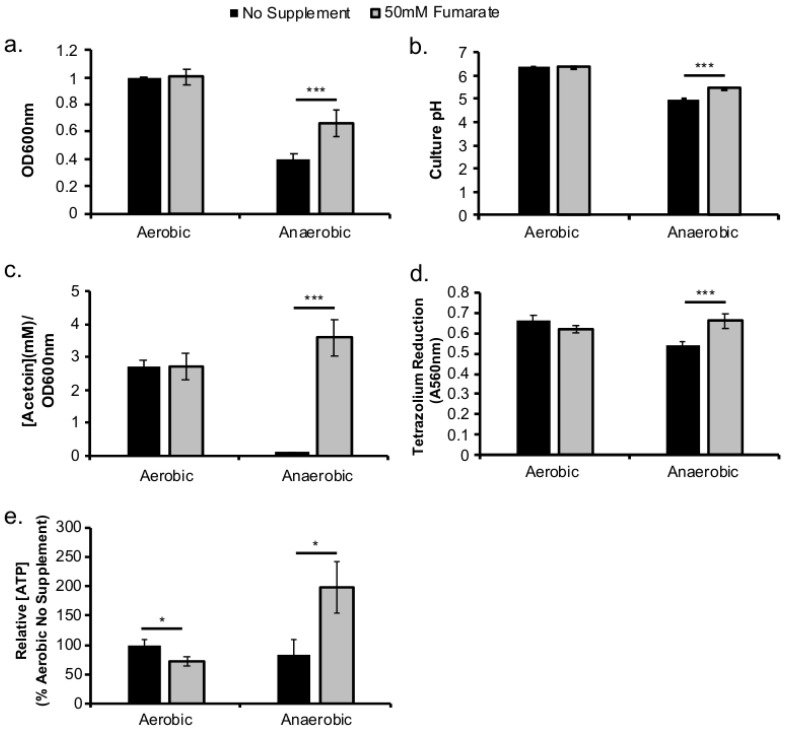
Fumarate supplementation results in enhanced *L. monocytogenes* anaerobic respiratory activity. *L. monocytogenes* was grown overnight (16–20 h) aerobically or anaerobically in BHI at 37 °C with or without supplementation of fumarate (50 mM). Culture optical density was measured at 600 nm (**a**). The pH (**b**) and acetoin concentration (**c**) were determined in culture supernatant. Reduction of tetrazolium salt (**d**) was determined in washed bacterial pellets. Relative intracellular ATP levels (**e**) was quantified using a luciferase-based assay. Averages of triplicates were plotted with error bars representing standard deviation. Data represent at least three independent experiments. Significant differences (*, *p* < 0.05; ***, *p* < 0.001) were calculated using the two-tailed Student’s *t*-test.

**Figure 2 pathogens-07-00096-f002:**
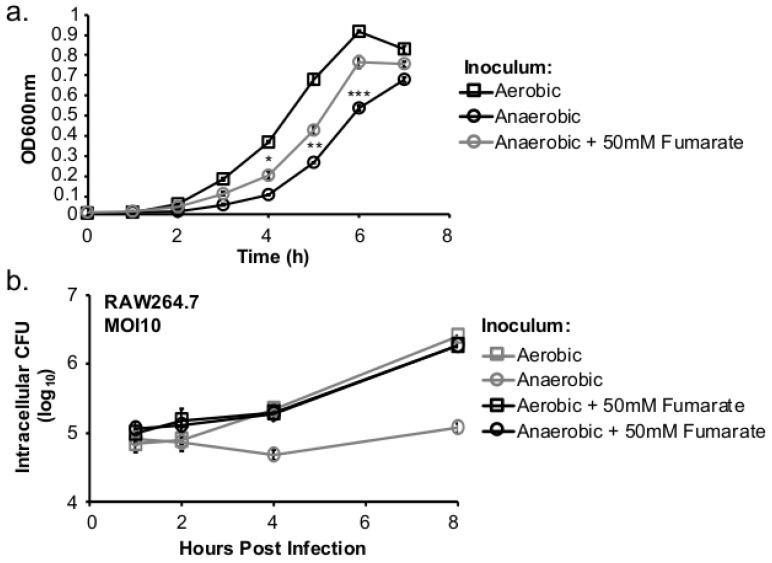
Fumarate supplementation results in enhanced transition from anaerobic to aerobic growth. Overnight cultures grown under aerobic or anaerobic conditions were used to inoculate fresh BHI media and grown under aerobic conditions (**a**) or to infect RAW264.7 macrophages for 30 min at an MOI of 10. (**b**) Averages of triplicates were plotted with error bars representing standard deviation. Data represent three independent experiments. Significant differences (*, *p* < 0.05; **, 0.001 < *p* < 0.01; ***, *p* < 0.001) were calculated between anaerobic inoculum and anaerobic, fumarate-treated inoculum using the two-tailed Student’s *t*-test.

**Figure 3 pathogens-07-00096-f003:**
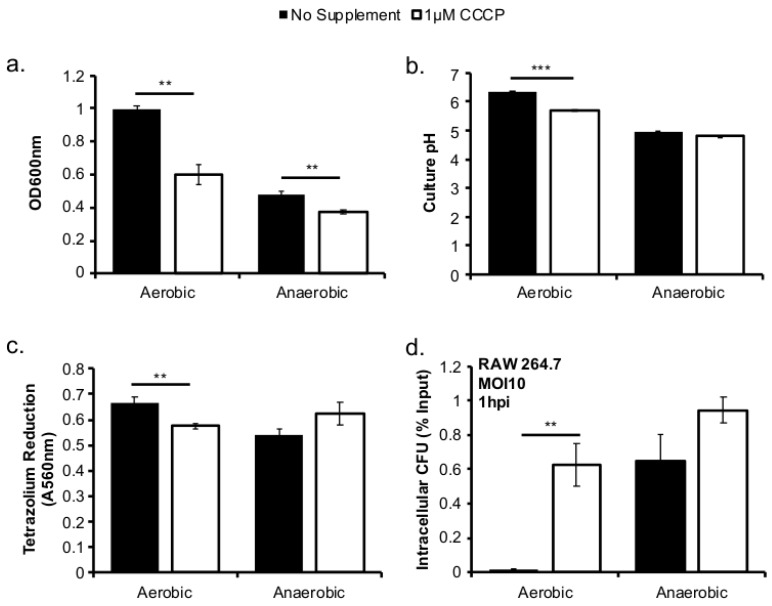
Inhibition of aerobic respiratory activity with CCCP phenocopies anaerobically grown *L. monocytogenes*. *L. monocytogenes* was grown at 37 °C in BHI with or without 1 µM CCCP. Culture optical density was measured at 600 nm. (**a**) pH was measured in culture supernatant. (**b**) While reduction of tetrazolium salts was measured in washed bacterial pellets. (**c**) Bacteria from overnight cultures were used to infect RAW264.7 macrophages for 30 min at an MOI of 10 where intracellular colony forming units (CFU) was determined and compared to the CFU in inoculum. (**d**). Averages of triplicates were plotted with error bars representing standard deviation. Data represent three independent experiments. Significant differences (**, 0.001 < *p* < 0.01; ***, *p* < 0.001) were calculated using the two-tailed Student’s *t*-test.

**Figure 4 pathogens-07-00096-f004:**
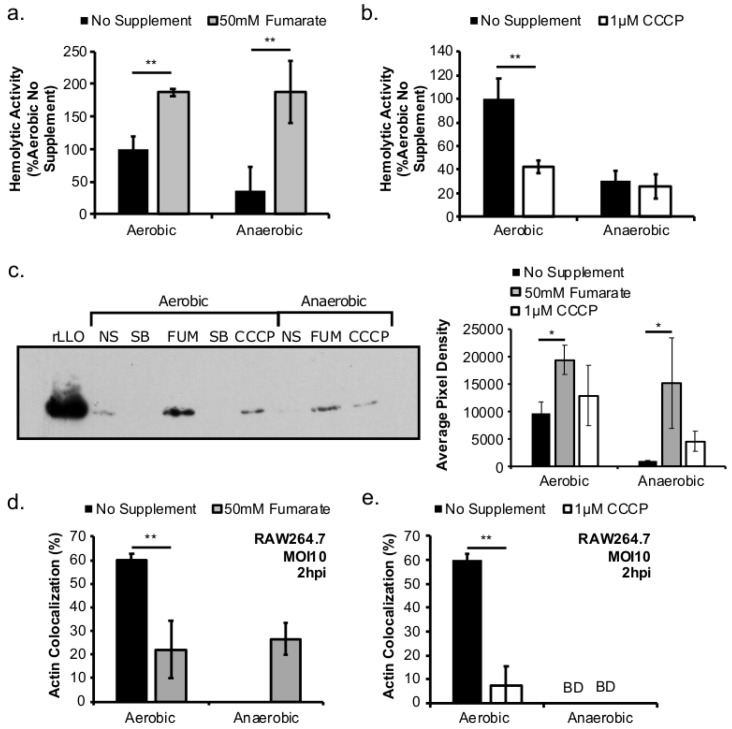
Modulations of respiratory activity alter production of LLO and actin co-localization. *L. monocytogenes* was grown overnight at 37 °C in BHI. Culture supernatant was used to quantify LLO activity using a hemolytic assay (**a**,**b**). Averages of triplicates were plotted with error bars representing standard deviation. Data represent three independent experiments. LLO abundance in culture supernatant was determined using immunoblotting (**c**, **left**) followed by pixel density analysis (**c**, **right**). Blot shown is representative of six independent experiments. (rLLO, recombinant LLO control; NS, no supplement; SB, sample buffer; FUM, fumarate-treated *L. monocytogenes*; CCCP, CCCP-treated *L. monocytogenes*.) Averages of pixel density from six blots were plotted with error bars representing standard error of the mean (n = 6). RAW264.7 macrophages were infected with overnight *L. monocytogenes* for 30 min at a MOI of 10. Actin colocalization was determined at 2 h post infection (hpi). BD, or below detection, indicates a lack of colocalization. Significant differences (*, *p* < 0.05; **, 0.001 < *p* < 0.01) were calculated using the two-tailed Student’s *t*-test.

**Figure 5 pathogens-07-00096-f005:**
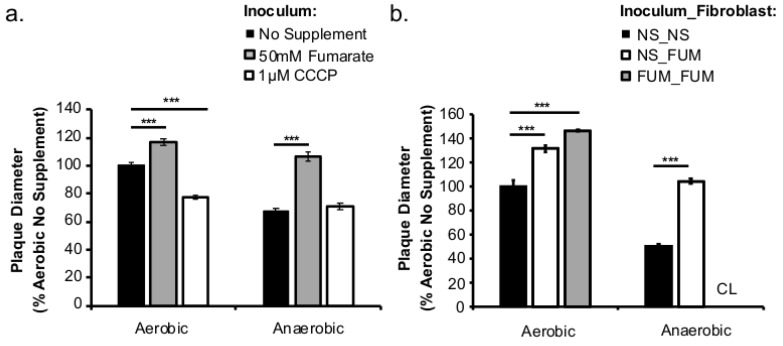
Prior fumarate treatments enhance cell-cell spread. Overnight cultures of *L. monocytogenes* were washed and used to infect fibroblast monolayer cells for 1 h. After 72 h of incubation, plaque sizes were measured by neutral red staining. Diameters of at least 30 plaques were measured. (**a**) Fumarate (50 mM) or CCCP (1 µM) was supplemented only in the growth of the bacterial inoculum. (**b**) Compared to no fumarate control (NS_NS), fumarate was added only to fibroblasts prior to infection (NS_FUM) or to both the growth of *L. monocytogenes* and fibroblasts prior to infection (FUM_FUM). No exogenous fumarate was added during infection. Fumarate supplementation in both *L. monocytogenes* and fibroblasts prior to infection resulted in complete lysis where plaque sizes were not determinable (CL, Complete Lysis). Significant differences (***, *p* < 0.001) were calculated using a single factor ANOVA analysis and error bars are representative of the standard error of the mean (SEM).

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
