# Peer review of "Stimulating Respiratory Activity Primes Anaerobically Grown Listeria monocytogenes for Subsequent Intracellular Infections"

_pathogens, 2018, doi:10.3390/pathogens7040096_

Round 1

Reviewer 1 Report

The research subject is worthy of investigation, fits within the scope of the journal and provides valuable information to the general reader of this journal. Overall, the manuscript is well-written, the experimental design is appropriate, methodologies utilized are sound and results provide novel insight into the mechanisms of pathogenesis in Listeria monocytogenes infection. My only suggestion is to change in the introduction (line 28) and within the whole manuscript the abbreviation of  "Listeria monocytogenes (Listeria)" in the classical "Listeria monocytogenes (L. monocytogenes)".

Author Response

Point 1: Change the abbreviation of "Listeria monocytogenes (Listera)" to "L. monocytogenes"

Response 1: We have corrected the abbreviation to "L. monocytogenes" throughout the manuscript and would like to thank the reviewer for the suggestion.

Reviewer 2 Report

The article written by Wallace et al. presents the results of a series of studies to demonstrate that in vitro anaerobically grown Listeria monocytogenes can provoke long-term infections in macrophages. In order to ascertain this, the authors performed carried out different culture conditions (aerobic and anaerobic, supplemented and not supplemented with fumarate or ATPase inhibitors) and then infected the eukaryotic cells quantifying different parameters to determine the metabolic, respiratory activity and the capacity of actin microfilament polymerisation once the infection occurred.

Results obtained indicated that fumarate supplementation significantly increases the intracellular survival of L. monocytogenes compared with untreated cells and the respiratory activity of the bacterial cells. Additionally, authors observed that LLO production and actin colocalisation differed in cells cultured in presence/absence of oxygen and with or no fumarate supplement.

GENERAL COMMENT

Overall, the manuscript is very well written, easy to follow and the scientific content is very relevant. Nevertheless, the text has several issues that need to be corrected and modified before being considered for publication. From my point of view, the weakest part is the methods section that in many cases lack on details of the protocols given if the assays are pretended to be replicated elsewhere. In other cases, the information of the assay is, simply, absent. Moreover, some other parts should be reorganised and the text as a whole must be formatted according to the journal standards. Please refer to the specific comments below in order to address the above mentioned issues.

ABSTRACT

L11 Microbiologically the term “Listeria” alone is not an elegant choice. Please use L. monocytogenes instead. Additionally, in other fragments use writing formulas such as “this microorganism” or similar. Please correct throughout the manuscript.

INTRODUCTION

L64-72 These lines belong to the Material and Methods, Results and Discussion section. Please remove and relocate accordingly.

RESULTS

This section should be divided in subsections in order to clarify the presentation of the outcomes obtained.

L74-76 This is a primary hypothesis that should be in the introductory section.

L86-89 This is discussion.

L112-115 Discussion.

L132-133 As displayed in figure 3, the growth reduction in aerobic culture is of approx. 0.4 U compared to the approx. 0.1 U in anaerobic growth. This difference should be stressed by the authors in this section.

L142-144 Discussion.

L161-163 This assay is not specified in the Materials and Methods section. Please add details of how the experiments were performed and how the results were quantified.

L171-174 Discussion.

L184-185 Discussion.

L208-209 This lines should be relocated in the Methods section.

L209-210 Discussion.

L216-220 Discussion.

DISCUSSION

L236 Authors should avoid using writing formulas such as “we and others”. Please rewrite and correct throughout the manuscript (e.g. in L244 use “it is reported” instead of “we reported”).

L295-297 A relevant reference should be added at the end of this sentence.

L297-301 As given, the information provided in these lines do not add any useful information for the purposes of the article since they are studies based on eukaryotic cellular activity and the consequences in multicellular organisms but not about bacterial pathogenesis. Please delete.

MATERIALS AND METHODS

The format of the subsections titles should be modified accordingly to the journal layout.

L319-320 Lot information is irrelevant. Please remove.

L326 specified the concentrations of fumarate used in the assays.

L335 Dulbecco’s Phosphate-buffered saline (DPBS).

L336-338 Please indicate the hours post infection in which the enumeration was performed.

L343 Phosphate-buffered saline (PBS)

L344 Display full name of MTT and provide details about the manufacturer.

L361 Dithiothreitol (DTT) – Manufacturer?

L379 Specify temperature of acetone.

L380 Which volume of sample was used to be resuspended in sample buffer’ Please be more specific when writing the methods followed.

L381-383 No details are given about how the immunoblotting was performed (transference to a membrane, volumes, times, temperatures of incubation, etc). Please check and modify the text accordingly.

L387 Please indicate washing solution used.

L388 DMEM?

L388-389 For the preparation of the sample the medium was not aspired first? Or the PFA was added directly to the well? This fragment is confusing for the reader.

L395-396 I guess that a fluorescence microscope was used for the quantification in this assay. Authors should give detailed information about how this observation was performed. Moreover, if images were taken for the quantification, they should be included in Figure 4d in order to facilitate the understanding of the results provided by the authors.

L401 How was the adjustment done? Please specify.

L410 Specify culture conditions.

L414 Please give details of neutral red staining solution and  volume of this solution used for cell staining.

An “Statistic analysis” should be added at the end of this section.

Additionally, a conclusion paragraph summarising the main findings and the significance and impact of the study carried out should be added at the end of this manuscript.

Round 2

Reviewer 2 Report

The authors have made a considerable effort to improve the manuscript and all the previous comments made have been properly assessed.

From my side, the paper is now ready for publication. Congratulations.